# Vitamin D as an Epigenetic Regulator: A Hypothetical Mechanism for Cancer Prevention via Inhibition of Oncogenic lncRNA HOTAIR

**DOI:** 10.3390/ijms26167997

**Published:** 2025-08-19

**Authors:** Samuel Trujano-Camacho, Ángel Pulido-Capiz, Victor García-González, Eduardo López-Urrutia, Carlos Pérez-Plasencia

**Affiliations:** 1Laboratorio de Genómica Funcional, Unidad de Biomedicina, Facultad de Estudios Superiores Iztacala, Universidad Nacional Autónoma de México, Tlalnepantla Estado de México 54090, Mexico; samuel.trujano1@gmail.com (S.T.-C.); e_urrutia@unam.mx (E.L.-U.); 2Experimental Biology PhD Program, DCBS, Universidad Autónoma Metropolitana-Iztapalapa, Mexico City 09340, Mexico; 3Departamento de Bioquímica, Facultad de Medicina Mexicali, Universidad Autónoma de Baja California, Mexicali 21100, Mexico; pulido.angel@uabc.edu.mx (Á.P.-C.); vgarcia62@uabc.edu.mx (V.G.-G.)

**Keywords:** cancer prevention, vitamin D, lncRNAs, cancer epigenetics, HOTAIR

## Abstract

Cancer remains a leading cause of mortality worldwide, arising from a complex interplay of genetic, epigenetic, and environmental factors. Although the role of micronutrients in cancer development has received limited attention, growing evidence suggests that vitamins, particularly vitamin D, may influence oncogenic pathways. This hypothesis manuscript explores the potential interaction between vitamin D and the oncogenic long non-coding RNA HOTAIR, providing a novel mechanistic explanation for the inverse correlation between vitamin D status and cancer risk. We support our hypothesis with in silico docking evidence, suggesting that vitamin D binds to bioactive domains within the structured regions of HOTAIR, potentially disrupting its interaction with chromatin regulators such as PRC2. This concept may offer a novel approach to cancer prevention and therapy.

## 1. Introduction

It is widely established that cancer is a multifactorial disease that still takes millions of lives every year [1]. Several lines of evidence support the involvement of both extrinsic and intrinsic factors in the dysregulation of the transcriptomic pattern that leads healthy cells to cancer development [2]. Studies of extrinsic factors are usually related to exposure to ionizing and non-ionizing radiation, pollutants in soil and water, and dietary choices such as high alcohol or sugar consumption [3]; however, less attention is paid to the relationship between micronutrients such as vitamins and cancer development [4].

Since diseases like beriberi and rickets ceased to be common due to population awareness and the availability of fortified food products, vitamins are taken for granted. Still, the widespread adoption of a Western-style diet in most countries drives the current increase in vitamin deficiency worldwide. Vitamins play substantial roles in cellular metabolism; consequently, their functions are dysregulated in cancer cells [5,6]. Several studies have shown correlations between vitamin levels and cancer risk, but the evidence is still contradictory, and further studies are needed before a vitamin can be established as a bona fide molecular marker or therapeutic target [7].

Vitamin D has two main forms: D2 (ergocalciferol) and D3 (cholecalciferol); the former is obtained from the diet, and the latter is produced in the skin through the photolysis of provitamin D3 [8]. Within the cell, both forms are hydroxylated to 1,25-dihydroxyvitamin D (1,25(OH)_2_D), which, in turn, binds the Vitamin D Receptor (VDR). This nuclear receptor regulates transcription by interacting with vitamin D response elements (VDREs) present in the promoter region of a plethora of genes involved in cellular functions such as adaptive immunity and cell differentiation [9].

Vitamin D deficiency has been linked to cancer as far back as the 1930s, when it was observed that internal cancer incidence decreased with higher UV exposure. Further studies linked 25(OH)D concentration and cancer risk [10]. It is now accepted that VDR-controlled genes regulate pathways central to cancer biology, including energy metabolism, apoptosis, and DNA repair [11,12]. This broad regulatory effect includes non-coding transcripts, mainly long non-coding RNAs (lncRNAs), which are transcribed from promoters containing VDR binding sites [13]. This relationship opens the possibility for one or more lncRNAs to be the effectors of the antioncogenic effect of vitamin D.

Essentially, lncRNAs exert their regulation through binding to other molecules, be they other nucleic acids or proteins. LncRNAs that bind miRNAs are referred to as miRNA sponges; lncRNAs that bind proteins can act as decoys when they bind competitively, or as guides and scaffolds when their binding favors the target’s function [14,15]. These regulatory functions are highly dependent on their specific sequences, subcellular localization, and structures [16]. These structural domains and functional motifs within each lncRNA enable the precise regulation of gene expression and the modulation of intracellular architecture [17].

A substantial number of lncRNAs have been implicated in the regulation of gene expression through their interaction with the Polycomb Repressive Complex 2 (PRC2). The activity of PRC2 is strongly influenced by intergenic sequences and structural elements within lncRNAs, many of which exhibit evolutionary conservation across species [18].

HOX transcript antisense intergenic RNA (HOTAIR) is an intergenic lncRNA that plays a pivotal role in various cellular processes through its interactions with the PRC2 and various transcription factors. Domain 1 of HOTAIR (nucleotides 212–300) is structurally composed of 11 helices, 8 terminal loops, and 3 junctions, enabling it to function as a scaffold for PRC2 and transcription factors. This structural configuration facilitates the recruitment of these regulatory proteins to transcription factor binding sites (TFBSs) on DNA, endowing HOTAIR with essential properties for orchestrating transcriptional regulation [19].

HOTAIR is one of the most extensively studied lncRNAs. It exerts broad regulation, affecting virtually every major signaling pathway in several cancer types [20], so its overexpression is closely linked to tumorigenesis and metastasis [21,22], while its underexpression modulates resistance to treatment [23]. These interactions are under deep scrutiny, mainly for therapeutic purposes [24]. For instance, ellipticine, a tetracyclic alkaloid, has been proven to directly bind HOTAIR, preventing it from guiding the chromatin remodeler EZH2 to increase BDNF transcription [25]. Furthermore, HOTAIR expression has been linked to that of VDR [26,27], suggesting that vitamin D modulates HOTAIR.

The 3D structure of HOTAIR appears to be well-defined and conserved across different cellular contexts [28,29], allowing for interactions with other molecules. We consider it highly unlikely that only exogenous small molecules, such as ellipticine, bind lncRNAs to regulate their function. So, we hypothesize that vitamin D regulation may occur directly. Consequently, sufficient levels of vitamin D could repress HOTAIR-driven tumorigenic pathways, offering a molecular basis for the observed inverse relationship between vitamin D status and cancer risk. To explore this hypothesis, we performed molecular docking simulations, which revealed that vitamin D3 can occupy putative binding pockets within the functional domains of HOTAIR. These pockets overlap with known PRC2 and transcription factor interaction sites. Our model suggests that vitamin D binding could sterically hinder or allosterically modulate HOTAIR’s interaction with chromatin regulators. Here, we present the first approach to proving our hypothesis, showing, for the first time, that vitamin D docks to bioactive sites within the HOTAIR structure in silico.

Thus, we propose a novel mechanism by which vitamin D may exert anticancer effects: through direct binding to the oncogenic lncRNA HOTAIR. This hypothesis bridges the physiology of vitamin D and the epigenetic regulation of cancer, offering a new perspective on its chemopreventive potential. Experimental validation via RNA pull-down, structural biology, and functional assays will be necessary to confirm this mechanism and evaluate its therapeutic implications.

## 2. Results

HOTAIR is overexpressed in multiple cancer types, predominantly in advanced clinical stages.

We first performed a bioinformatic analysis using two databases containing TCGA-enriched samples, which revealed that HOTAIR is significantly overexpressed in at least 15 different tumor types compared to non-tumor controls (Figure 1A). This finding highlights the potential role of HOTAIR as a master regulatory lncRNA in a broad spectrum of cancers.

Then, we analyzed HOTAIR expression by stage (1–4) in the main cancer types, in terms of incidence and mortality, using GLOBOCAN data. This evidence supports the role of HOTAIR as a regulator of various tumor types, highlighting its involvement in the most advanced stages (3 and 4) (Figure 1B–G). Thus, HOTAIR contributes to tumor progression, particularly through mechanisms related to invasion and metastasis.

Molecular Docking Analysis of Cholecalciferol Binding to Domain 1 of HOTAIR

To understand the potential relationship between the HOTAIR Domain 1 and vitamin D_3_ (cholecalciferol), molecular docking was performed using the MOE platform. The AC1NOD4Q compound, previously reported to have a specific affinity for HOTAIR, was used as a positive control (Figure 2 and Figure 3).

## 3. Discussion

The role of HOTAIR in regulating the development of various tumors has been extensively investigated. In particular, its interaction with the PRC2 complex has been identified as a key mechanism mediated by Domain 1 of HOTAIR. This interaction promotes the deposition of the H3K27Me3 mark in DNA regions enriched with polypurine (GA) sequences or HOTAIR-specific motifs. As a result, PRC2 is recruited to over 800 genomic regions distributed across multiple chromosomes, primarily located within promoter and intronic regions. These findings underscore HOTAIR’s function as a multiprotein scaffold capable of specific interactions with both DNA and protein components through its structured domains [30]. The importance of identifying HOTAIR based on the specificity of its domains has prompted the development of drugs designed to inhibit it from a structural perspective. AC1NOD4Q, the compound used as positive control in our study, effectively disrupted the interaction between EZH2 and HOTAIR. As a result, the activity of PRC2 on specific gene targets associated with this axis was reduced [31].

HOTAIR plays a central role in cancer progression by orchestrating multiple oncogenic processes beyond mere gene silencing. Its overexpression rewires transcriptional programs that promote epithelial-to-mesenchymal transition (EMT), a critical step in invasion and metastasis [32]. HOTAIR activates EMT by stabilizing transcriptional repressors of E-cadherin and upregulating mesenchymal markers, facilitating cell motility and dissemination. Additionally, it modulates key signaling pathways, including Wnt/β-catenin and PI3K/AKT, amplifying proliferative and anti-apoptotic signals in various tumors (reviewed in [33]).

Clinically, HOTAIR drives drug resistance by repressing genes involved in apoptosis and altering the expression of genes related to drug metabolism. In breast and colorectal cancers, its upregulation is directly linked to poor therapeutic response and higher recurrence rates [34,35]. HOTAIR also serves as a molecular sponge for tumor-suppressive miRNAs, releasing oncogenic mRNA targets from repression and amplifying oncogene expression; miR-301a-3p, miR-1277-5p, and miR-129-5p in glioblastoma, colorectal cancer, and breast cancer, respectively, are examples of this [36,37,38]. Its tertiary structure forms well-defined domains that facilitate interactions with a range of cancer-promoting proteins, making it a scaffold for malignancy [39].

These multifaceted roles establish HOTAIR not just as a biomarker but as a functional driver of cancer aggressiveness. Its targeting, whether by antisense oligonucleotides or small molecules, represents a promising therapeutic avenue in tumors where current treatments fall short.

To establish a relationship in this context, it has been demonstrated that vitamin D_3_ upregulates CYP24A1 in both osteoblasts and hepatocytes, which in turn induces EZH2 expression. This leads to the transcriptional repression of genes such as DKK, which inhibit several well-established oncogenic pathways, including the Wnt signaling pathway [40,41]. This suggests that the functions of PRC2 in maintaining cellular homeostasis involve the proper regulation of transcription factor patterns in a spatiotemporal manner. Alterations such as the overexpression of lncRNAs can interfere with this regulation, leading to specific deregulations that promote tumor development [42]. In this way, vitamin D would promote adequate homeostasis, as indicated by the clinicopathological relationships that link deficiencies of this vitamin in circulation to an increased risk of cancer development [43]. The evidence suggests that, when processed, vitamin D favors the expression of EZH2 and the orderly functioning of PRC2. However, in tumor conditions, the high expression of HOTAIR leads to the oncogenic functioning of this complex [44].

This allows us to hypothesize that two concurrent or sequential events promote tumor development. The first is a deficiency in circulating vitamin D, which has been widely linked to an increased risk of tumor formation. The second is an elevated HOTAIR expression. We propose that vitamin D binds directly to the oncogenic lncRNA HOTAIR; therefore, its apparent overexpression may be a consequence of lower vitamin D concentrations releasing HOTAIR from its regulatory control, allowing it to increase epigenetic alterations through the HOTAIR/PRC2 axis. Consequently, these changes may be reversible upon restoration of adequate vitamin D levels, potentially explaining the observed increases in chemosensitivity across various tumors [45].

The strong experimental validation needed to confirm these findings is highly plausible due to the substantial advances in our understanding of the interactions between RNA and small molecules [46]. The interaction between a vitamin and an ncRNA opens the possibility of an additional regulatory network underpinning cellular metabolism.

## 4. Materials and Methods

Bioinformatic Profiling of HOTAIR Expression in Human Cancer.

To assess HOTAIR expression levels across various cancer types, two bioinformatics platforms integrating The Cancer Genome Atlas (TCGA) data were employed. HOTAIR expression between tumor and adjacent normal tissues across 32 cancer types was compared in the Tumor Immune Estimation Resource (TIMER). Expression levels were normalized and presented as log_2_ transcripts per million (TPM) [47]. To validate these results, HOTAIR expression was further analyzed by cancer stage in six major cancer types using the UALCAN Cancer Data Analysis Portal, developed by the University of Alabama at Birmingham [48]. Statistical significance for both datasets was determined using the Wilcoxon test. Significance codes are as follows: *** < 0.001 ≤ ** < 0.01 ≤ * < 0.05.

Molecular Modeling and Docking of HOTAIR and Cholecalciferol.

The three-dimensional (3D) structure of the HOTAIR domains was initially predicted using the trRosettaRNA tool, a deep learning–based platform specifically designed for RNA tertiary structure modeling. trRosettaRNA utilizes transformer-based neural networks to predict pairwise nucleotide geometries, including distances, orientations, and torsion angles. These predicted constraints are then used to guide the generation of physically plausible 3D RNA models. When necessary, Rosetta energy minimization is applied to resolve steric conflicts and refine geometry. For this study, the RNA sequence corresponding to the EZH2-binding region (nucleotides 212–300, Domain 1) of HOTAIR was submitted to trRosettaRNA, and the structure with the highest internal confidence score was selected for further analysis.

The selected model of Domain 1 was then imported into the Molecular Operating Environment (MOE) software (Chemical Computing Group, Montreal, QC, Canada) for structural preparation. An initial geometric optimization was performed using the Amber10:EHT force field to refine the RNA structure [47]. The 3D structure of cholecalciferol was retrieved from PubChem (https://pubchem.ncbi.nlm.nih.gov/ accessed on 14 May 2025) in SDF format and imported into MOE. To explore ligand flexibility, multiple conformations of cholecalciferol were generated using the Conformational Search module.

For molecular docking, a blind docking strategy was employed, allowing the ligand to explore the entire molecular surface of the RNA without predefined binding site constraints. The Triangular Matcher algorithm was used to generate the initial ligand poses, followed by refinement using the GBVI/WSA dG scoring function. Binding affinities were estimated using the London dG method. A total of 30 poses were generated per ligand, and the top 5 poses with the lowest S scores (in kcal/mol) were retained for interaction analysis. All selected poses underwent post-docking energy minimization to resolve steric clashes and optimize binding geometry.

The RNA structure was treated as semi-rigid, while ligands were modeled with full conformational flexibility. Molecular interaction analyses focused on identifying hydrogen bonds, hydrophobic contacts, and π–π stacking interactions between HOTAIR Domain 1 and cholecalciferol, using MOE’s 3D visualization tools

To evaluate docking specificity, AC1NOD4Q, a compound previously validated as a HOTAIR-binding molecule, was used as a positive control. As a negative control, vitamin C (ascorbic acid), which is a small, highly polar, and hydrophilic molecule with no reported RNA-binding activity and is structurally unrelated to lipophilic ligands like vitamin D, was used. Using the same blind docking protocol in MOE, as applied to the test compounds, ascorbic acid failed to generate any stable binding poses within the HOTAIR Domain 1 structure.

## Figures and Tables

**Figure 1 ijms-26-07997-f001:**
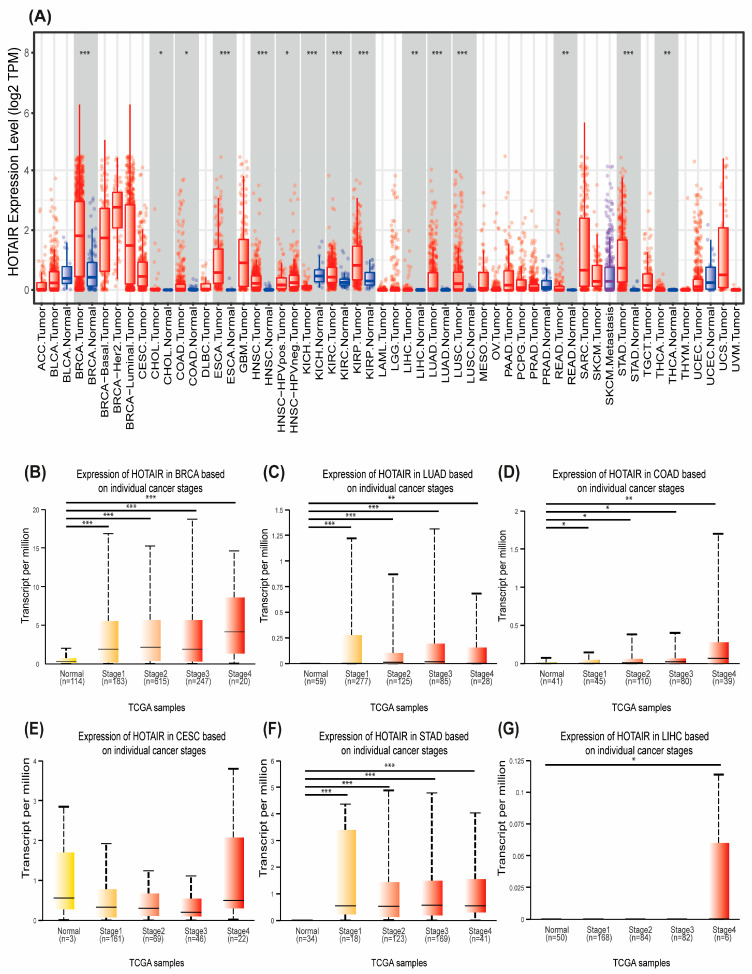
Differential expression of HOTAIR across cancer types and stages using two bioinformatic tools. (**A**) HOTAIR expression levels across 32 cancer types, as analyzed using the TIMER platform. (**B**–**G**) Stage-specific HOTAIR expression levels in breast (BRCA), lung adenocarcinoma (LUAD), colon (COAD), cervical (CESC), stomach (STAD), and liver (LIHC) cancers, obtained from the UALCAN portal. Statistical significance: *** *p* < 0.001, ** *p* < 0.01, * *p* < 0.05.

**Figure 2 ijms-26-07997-f002:**
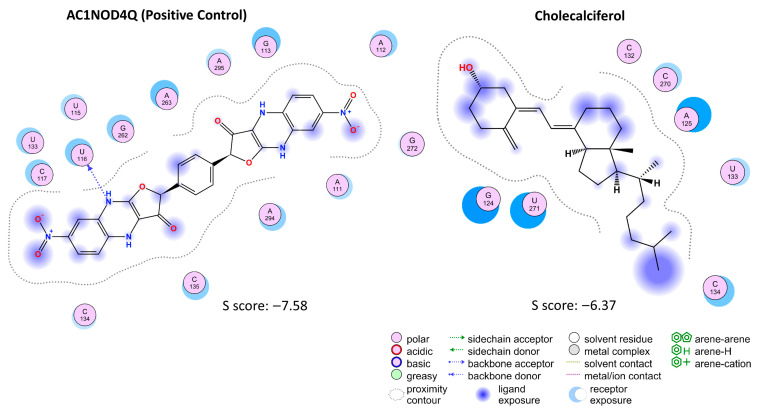
Schematic representation of molecular interactions between HOTAIR Domain 1 and Ligands AC1NOD4Q (positive control) and cholecalciferol. Two-dimensional interaction maps illustrate the key binding features for each complex, highlighting hydrophobic regions (blue-shaded areas), hydrogen bonds (dotted lines), and proximal RNA residues. Docking S score values (kcal/mol) indicate estimated binding affinities, with more negative scores reflecting stronger and more stable interactions.

**Figure 3 ijms-26-07997-f003:**
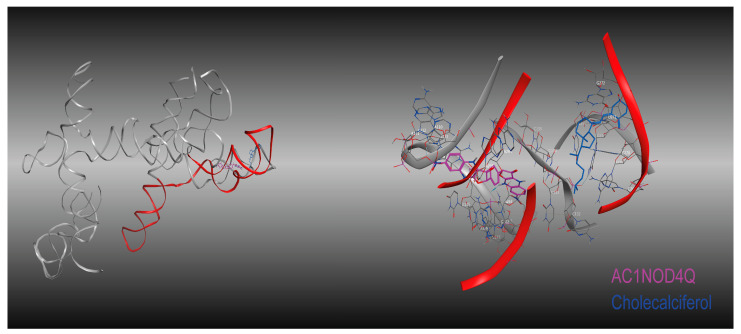
Spatial positioning of the ligands AC1NOD4Q (Positive Control) and cholecalciferol within HOTAIR Domain 1. The region of HOTAIR involved in interaction with EZH2 is highlighted in red. Ligand AC1NOD4Q is shown in purple, while cholecalciferol is depicted in blue, illustrating their respective binding sites relative to this functional domain. The right panel presents a magnified view highlighting the interactions between specific nucleotides and cholecalciferol -shown in blue-, alongside the positive control AC1NODQ4 -shown in pink-.

## Data Availability

Data available on request.

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
