# Peer review of "Vitamin D as an Epigenetic Regulator: A Hypothetical Mechanism for Cancer Prevention via Inhibition of Oncogenic lncRNA HOTAIR"

_ijms, 2025, doi:10.3390/ijms26167997_

Round 1
Reviewer 1 Report
Comments and Suggestions for Authors
This manuscript investigates a novel and very interesting hypothesis that vitamin D3 binds directly to the oncogenic lncRNA HOTAIR and modulate its function. The authors clearly explain this hypothesis and provide evidence for it through TCGA-based expression analysis and molecular docking simulations. The manuscript is overall very well written, with a logical structure and an engaging story. Here are several suggestions that could improve reproducibility, and biological depth.
- Transparency and reproducibility would be improved by adding an additional table (as a supplementary material) that summarizes the TCGA datasets (tumor types, sample sizes and any inclusion or exclusion criteria).
- The authors should present a stage-specific HOTAIR expression analysis (Stages I–IV) in a separate table and potentialy analyze its correlation with VDR expression TCGA data to support the suggested regulatory axis.
- HOTAIR's role as a trans-acting miRNA sponge should be discussed in more depth, as this further supports its potential as a therapeutic target (Ren et al. 2019, 11:29; Clin Epigenetics).
- It would also be beneficial to go into more detail about HOTAIR's function as a molecular scaffold, specifically how its structured domains mediate simultaneous interactions with transcription factors such as VDR and chromatin remodelers like PRC2.
- The authors could also discuss potential differences in HOTAIR expression between HPV-positive and HPV-negative tumors, particularly in HNSCC and cervical carcinoma, where HPV status signicifantly alters the immune landscape and prognosis - but in opposite directions. Additionally, at least in supplemental material, stage-specific HOTAIR expression levels in HNSCC, and/or HPV-positive and HPV-negative HNSCC, would be beneficial to be presented.
Author Response
This manuscript investigates a novel and very interesting hypothesis that vitamin D3 binds directly to the oncogenic lncRNA HOTAIR and modulate its function. The authors clearly explain this hypothesis and provide evidence for it through TCGA-based expression analysis and molecular docking simulations. The manuscript is overall very well written, with a logical structure and an engaging story. Here are several suggestions that could improve reproducibility, and biological depth.
1. Transparency and reproducibility would be improved by adding an additional table (as a supplementary material) that summarizes the TCGA datasets (tumor types, sample sizes and any inclusion or exclusion criteria).
R. Dear revisor, thanks for the correction. The figures with TCGA datasets were extracted from the TIMER and UALCAN databases, as described in the Methods section. We have also added citations for each database. While the respective papers describe the tumor types, sample sizes, and inclusion/exclusion criteria, in section Methods, subsection “Bioinformatic Profiling of HOTAIR Expression in Human Cancer” we have summarized the information to demonstrate the importance of HOTAIR in most tumors.
2. The authors should present a stage-specific HOTAIR expression analysis (Stages I–IV) in a separate table and potentially analyze its correlation with VDR expression TCGA data to support the suggested regulatory axis.
R. Dear reviewer despite being a very relevant and interesting question in another scientific context. The main objective of this Hypothesis manuscript is to show that Vitamin D, directly regulates the bioactive domains of HOTAIR. We do not suggest an axis between HOTAIR and VDR. We propose that an inverse correlation between serum levels of Vitamin D and the biological activity of HOTAIR could lead to initiation and progression of cancer.
HOTAIR's role as a trans-acting miRNA sponge should be discussed in more depth, as this further supports its potential as a therapeutic target (Ren et al. 2019, 11:29; Clin Epigenetics).
R. Thank you for your kind suggestion; we discussed this in lines 165–170 and 184–185.
3. It would also be beneficial to go into more detail about HOTAIR's function as a molecular scaffold, specifically how its structured domains mediate simultaneous interactions with transcription factors such as VDR and chromatin remodelers like PRC2.
R. Dear reviewer, we have deeply analyzed and discussed the HOTAIR function as a molecular scaffold; particularly with PRC2. Lines 71-83; 157-170; 196-205 and 209-214. However there are not information about the interaction between HOTAIR and VDR.
4. The authors could also discuss potential differences in HOTAIR expression between HPV-positive and HPV-negative tumors, particularly in HNSCC and cervical carcinoma, where HPV status signicifantly alters the immune landscape and prognosis - but in opposite directions. Additionally, at least in supplemental material, stage-specific HOTAIR expression levels in HNSCC, and/or HPV-positive and HPV-negative HNSCC, would be beneficial to be presented.
R. We respectfully disagree. The primary objective of this study was to examine the hypothetical role of vitamin D as an inhibitor of HOTAIR bioactive sites, drawing on evidence from molecular docking experiments and the potential inverse association between serum vitamin D levels and cancer incidence. While we acknowledge that HPV may contribute to the activation of oncogenic pathways within this complex biological context, we believe that the interplay among HPV, HOTAIR, and vitamin D deserves separate investigation. Such an analysis would be more appropriately addressed in a dedicated study specifically focused on the role of HPV and HOTAIR in cancer development.
Reviewer 2 Report
Comments and Suggestions for Authors
This manuscript my Trujano-Camacho et al presents an interesting hypothesis that vitamin D directly binds to HOTAIR and disrupts its oncogenic functions. The manuscript provides a intriguing synthesis of literature linking vitamin D deficiency to cancer, the oncogenic role of HOTAIR, and the feasibility of RNA–small molecule interactions.
Comments:
1. The docking methods, including the parameters, scoring criteria, RNA flexibility consideration, and validation metrics, should be described in detail.
- Figures 2 and 3 lack resolution and detailed annotation. Docking poses are not clearly labeled, and interaction types (e.g., H-bonds, hydrophobic interactions) are vaguely described.
- The text labels in Figure 1 are too small to be legible.
- While a positive control is included in the docking analysis, a negative control should also be included to assess whether vitamin D binding is non-specific or not.
Author Response
This manuscript my Trujano-Camacho et al presents an interesting hypothesis that vitamin D directly binds to HOTAIR and disrupts its oncogenic functions. The manuscript provides a intriguing synthesis of literature linking vitamin D deficiency to cancer, the oncogenic role of HOTAIR, and the feasibility of RNA–small molecule interactions.
Comments:
1. The docking methods, including the parameters, scoring criteria, RNA flexibility consideration, and validation metrics, should be described in detail.
R.- Dear reviewer, thank you very much for your positive feedback; we have added a complete section in Methods that you can review in lines 232-263
2. Figures 2 and 3 lack resolution and detailed annotation. Docking poses are not clearly labeled, and interaction types (e.g., H-bonds, hydrophobic interactions) are vaguely described.
R.- The Figures have been corrected as suggested.
3. The text labels in Figure 1 are too small to be legible.
R. Dear revisor, the figure has now been corrected.
4. While a positive control is included in the docking analysis, a negative control should also be included to assess whether vitamin D binding is non-specific or not.
R. We thank for the suggestion. In response, ascorbic acid (vitamin C) was included as a negative control in the docking experiments described in lines 265-271. Ascorbic acid is a small, highly polar, and hydrophilic molecule with no reported RNA-binding activity and is structurally unrelated to lipophilic ligands like vitamin D. Using the same blind docking protocol in MOE, as applied to the test compounds, ascorbic acid failed to generate any stable binding poses within the HOTAIR Domain 1 structure.
Round 2
Reviewer 2 Report
Comments and Suggestions for Authors
The authors have addressed all my comments.